# The Relationship between Daily Concentration of Fine Particulate Matter in Ambient Air and Exacerbation of Respiratory Diseases in Silesian Agglomeration, Poland

**DOI:** 10.3390/ijerph16071131

**Published:** 2019-03-29

**Authors:** Małgorzata Kowalska, Michał Skrzypek, Michał Kowalski, Josef Cyrys, Niewiadomska Ewa, Elżbieta Czech

**Affiliations:** 1Department of Epidemiology, School of Medicine in Katowice, Medical University of Silesia, 40-055 Katowice, Poland; mkowalska@sum.edu.pl; 2Department of Biostatistics, School of Public Health in Bytom, Medical University of Silesia, 40-055 Katowice, Poland; mskrzypek@sum.edu.pl (M.S.); eniewiadomska@sum.edu.pl (N.E.); emczech@sum.edu.pl (E.C.); 3Environmental Exposure Assessment Group, Institute of Epidemiology, Helmholtz Zentrum München, 85764 Neuherberg, Germany; cyrys@helmholtz-muenchen.de

**Keywords:** PM_10_, PM_2.5_, exposure, respiratory diseases

## Abstract

The relationship between the worsening of air quality during the colder season of the year and respiratory health problems among the exposed population in many countries located in cold climates has been well documented in numerous studies. Silesian Voivodeship, a region located in southern Poland, is one of the most polluted regions in Europe. The aim of this study was to assess the relationship between daily concentration of particulate matter (PM: PM_2.5_ and PM_10_) in ambient air and exacerbations of respiratory diseases during the period from 1 January 2016 to 31 August 2017 in the central agglomeration area of Silesian Voivodeship. The study results confirmed a significant increase of daily fine particulate matter concentration in ambient air during the cold season in Silesian Voivodeship with a simultaneous increase of the number of outpatient visits and hospitalizations due to respiratory diseases. The moving average concentration was better suited for the modelling of biological response as a result of PM_2.5_ or PM_10_ exposure than the temporal lag of health effects. Each increase of dose expressed in the form of moving average concentration over a longer time leads to an increase in the daily number of respiratory effects. The highest risk of hospitalization due to respiratory diseases was related to longer exposure of PM expressed by two to four weeks of exposure; outpatient visits was related to a shorter exposure duration of 3 days.

## 1. Introduction

The monitoring of air quality in Poland over the past two years indicated the occurrence of two winter smog episodes, with the first occurring during the period between 4 January and 8 January 2016 and the second between 7 January and 11 January 2017 [1]. Both of these winter smog episodes were an important public health issue which induced a great concern among residents, particularly the inhabitants of Silesian Voivodeship, an urbanized, historical, and coal mining industrial region located in southern Poland with over 3.8 million people [2]. Therefore, it is justifiable to estimate the actual relationship between the rapidly worsening air quality during winter smog episodes and the exacerbation of respiratory diseases among the exposed population. The obtained results are necessary for the environmental health risk communication and will prove to be a useful indicator for the implementation of the necessary health policy actions in the region. Episodes of winter smog in Poland occur every year, especially during the colder winter season. The level of environmental health perception in the country is understandably categorized as one of the lowest in Europe. However, contemporary media outlets have publicized and sensationalized the problem, leading to widespread panic among inhabitants. The situation is particularly difficult in the Silesia, a region where coal is mined and many people are employed in the mining industry. Our task is to document the relationship between the emerging hazard and the health of the population and communicate the risk based on reliable evidence. We believe that the results of our study will help to improve the inhabitants’ knowledge on the real hazard, and will reinforce the social activity needed to improve the quality of the environment. Among the well-described pulmonary adverse health effects associated with air pollution are short-term effects such as inflammatory reactions of the lungs and bronchi, and respiratory symptoms (coughing, wheezing, and problems with breathing). Additionally, increases in daily hospitalization, respiratory mortality, and increases in the usage of medications were reported [3]. The available data point to a significant increase in sales of medicaments used to treat exacerbations of respiratory diseases in the cold season, with a maximum in January and February [4]. This is similar to other countries, where a relationship between the concentration of fine particulate matter (PM_2.5_) and daily mortality due to respiratory diseases were observed [5,6,7]. Typically in environmental epidemiology, the moving average or temporal lags are used as a measure of exposure during the estimation of the concentration–response effects [8,9].

The aim of this study was to assess the relationship between daily concentrations of fine particulate matter in ambient air and exacerbations of respiratory diseases registered in the period from 1 January 2016 to 31 August 2017 in the central agglomeration area (CAA) of the Silesian Voivodeship, a southern region in Poland.

## 2. Materials and Methods 

Secondary epidemiological data of daily registered outpatient visits or hospitalizations of Silesian inhabitants due to total respiratory diseases (J00–J99 by ICD-10) in the study period (1 January 2016 to 31 August 2017) were obtained from the National Health Fund database in Katowice. The original data included the entire Silesian Voivodeship, however, the final database was limited to the central agglomeration area (CAA) with a population of 2,429,471 inhabitants (2015 year) and an area of 3337 km^2^. The total number of outpatient visits due to respiratory diseases in the study period was 3,550,901 and the total number hospitalizations was 60,346. CAA included the Upper Silesia Agglomeration and the central part of the Silesian zone with the following powiats: Będzin, Zawiercie, Tarnowskie Góry, Gliwice, and Bieruń-Lędziny counties. This decision was a consequence of the need to maintain consistency with the health data on air quality and to relate it with the real existing definition of CAA in the previous system of prognosis of the air quality index. It was defined at a website which existed until 1 January 2019. Figure 1 presents the location of CAA in the Silesian Voivodeship.

Air pollution and meteorological data in the study period, including daily concentration of SO_2_, NO_2_, NO_x_, ozone, and CO in ambient air, as well as daily temperature, relative humidity, and atmospheric pressure, were gathered from the Provincial Inspectorate of Environmental Protection in Katowice database [1]. Daily average concentrations of PM_2.5_ and PM_10_ (particles with an aerodynamic diameter less or equal 2.5 and 10 micrometers, respectively) were available only from two automatic measuring stations (urban background): in Katowice at Kossutha str. and in Gliwice at Gallus str. (out of all 15 stations located in the region). Because of very high correlation coefficients between daily averages recorded at both stations (*r* = 0.93, *r* = 0.90, and *r* = 0.99, for PM_2.5_, SO_2_, and ambient air daily air temperature or relative air humidity, respectively) values from Katowice station were used in the further analysis of the measurements. Moreover, a similarly high correlation was obtained for the other pollutants measured in Katowice and in other stations located in the study region (CAA) and in characterizing the agglomeration of the urban background [4].

To assess the relationship between ambient air pollution and the number of outpatient visits or hospitalizations due to respiratory diseases, the multivariable log-linear Poisson regression model was used. The model was linked with Equation (1): log[E(ND)] = Xβ,(1)where E(ND) is the observed daily number of outpatient visits or hospitalizations (dependent variable), X is a vector of independent variables (daily concentration of particulate matter, daily meteorological conditions, the season of the year, influenza episodes, and weekend days), and β means the calculated regression coefficient. The variables describing ambient air pollutant were expressed first as 1, 3, 5, 7, 14, and 30-day moving averages of PM_10_ and PM_2.5_ concentrations. Next, the exposure was expressed as a temporal lag of 1, 3, 5, 7, 14, and 30 days of outpatient visits or hospitalization. The confounding factors like daily average air temperature, relative humidity, atmospheric pressure, season (according to astronomical seasons), and week days (working day vs. holiday), but only for outpatient visits, were included in the model. Moreover, there was no observed influenza epidemic in the study period. Results of the model were presented by the relative risk (RR) of respiratory effect related to the increase in concentration of PM_10_ or PM_2.5_ by interquartile range (IQR) and were calculated using Equation (2):RR = exp (β × delta),(2)where β is the regression coefficient and delta is the IQR. The level of statistical significance used was α = 0.05. Calculations were conducted using SAS version 9.4 (SAS Institute Inc., Cary, North Carolina, USA).

## 3. Results

The study results confirmed the significant worsening of the ambient air quality during cold seasons in the study region (central area of Silesian Voivodeship, Poland), with the occurrence of two winter smog episodes in the study period. Both of the smog episodes were in January—the first one occurred from 4 January 2016 to 7 January 2016 and the second one from 7 January 2017 to 11 January 2017. In the second period, the temperature of ambient air was very low and the concentration of particulate matter was the highest. Median and IQR values for PM_2.5_ and PM_10_ concentrations in winter time were 44 (49.0) µg/m^3^ and 52.58 (53.22) µg/m^3^, respectively. 

Figure 2 shows the daily mean concentration of PM_2.5_ and PM_10_ measured in the study region between 1 January 2016 and 31 August 2017. The highest concentration of fine particulate matter was observed in the winter season of 2016/2017. In the time period from January 2017 until March 2017 the daily mean concentrations of PM_10_ were higher than 150 µg/m^3^ on 17 days, and daily mean concentrations of PM_2.5_ were higher than 100 µg/m^3^ on 20 days. 

Table 1 shows the seasonal variability of air pollutants as well as the daily mean of the health outcomes occurring (hospitalization, outpatient visits) stratified for different seasons. The highest number of registered outpatient visits and hospitalizations due to respiratory diseases (codes J00–J99 according to ICD-10) was also observed in the cold season of 2016 and 2016/2017, with median values of 9664.5 (10,746.5) and 136 (69.5), respectively.

In Table 2, the Spearman correlation coefficients between air pollutants and meteorological conditions measured at the two monitoring sites in Katowice and Gliwice are shown. A strong positive and statistically significant correlation between PM_10_ and PM_2.5_ concentration and other measured pollutants such as NO_2_, NO_x_, and CO was observed. A negative and also statistically significant correlation was noted for particulate matters and temperature, wind speed, or ozone concentrations. Higher values of relative humidity significantly increased the concentration of fine particles (PM_2.5_), whereas the effect in the case of PM_10_ was a bit lower. Moreover, the dominant direction of the wind was the west direction (49.5%), followed by the south (20.5%) and east (20.5%) direction.

Figure 3 and Figure 4 illustrate the results of the multivariable analysis assessing the risk ratio of daily outpatient visits (A) or hospitalization (B) due to respiratory disorders related to the increase of fine particulate matter concentrations by IQR value (24.5 µg/m^3^ for PM_10_ and 22.5 µg/m^3^ for PM_2.5_) in two scenarios of exposure: expressed by the moving average concentration or related to a temporal lag of health effect. 

It is interesting that statistically significant increases of health effects (both outpatient visits and hospitalizations) were observed in the case of dose expressed by two-week moving average concentrations. Despite the fact that we observed an increase in respiratory health effects related to an increase in particulate matter concentration by IQR, the pattern is a bit different for outpatient visits than in the case of hospitalizations. It was noticed that the 14-day moving average concentration of particulate matter was responsible for the highest risk of outpatient visits, RR = 1.049 (95%CI: 1.017–1.084) for PM_2.5_ and RR = 1.053 (95%CI: 1.022–1.086) for PM_10_, respectively. Moreover, both values were statistically significant. A similar observation was also seen in hospitalizations and 14-day moving average concentrations of PM_10_, however, the highest risk was observed for 30-day moving average concentrations of pollution, RR = 1.054 (95%CI: 1.009–1.099) for PM_2.5_ and RR = 1.053 (95%CI: 1.01–1.097) for PM_10_. In the case of the moving average concentration of PM_10_, a more evident respiratory effect was related with longer time of exposure and a statistically significant effect was obtained for 14-day moving average concentrations for both outpatient visits and hospitalizations. A similar relation was observed for PM_2.5_; the greatest and statistically significant risk was observed for two-week moving average concentrations in the case of outpatient visits as well as hospitalizations. In conclusion, we have to underline that the obtained risk ratios were small, as an increase of PM_10_ or PM_2.5_ concentration by IQR value increased the risk of outpatient visits or hospitalizations due to respiratory diseases by, on average, 2–5%.

The pattern of concentration–response effect in the case of exposure expressed by the temporal lag of health effects in relation to measuring day is a bit different. In the case of outpatient visits, the highest and statistically significant risk was observed with three or even five days lag after exposure for both pollutants, PM_2.5_ and PM_10_. In the case of hospitalizations, significant health effect was noted only for two-week lags of exposure. The highest risks related with PM_2.5_ were as follows: RR = 1.024 (95%CI: 1.004–1.044) for 14-day lags in hospitalizations and RR = 1.027 (95%CI: 1.01–1.043) for 3-day lags in the case of outpatient visits.

## 4. Discussion

The highest levels of air pollutants in the CAA occurred during the winter season (especially in January 2016 and 2017). This observation is similar to those reported from other countries, where the worst quality of ambient air is found in the cold season [10,11]. A major source of PM emission in winter in Poland remains outdated individual heating stoves in which people burn poor quality coal, biomass, or even garbage [4,12]. Numerous educational campaigns that have been raised for many years in an effort to improve the risk perception of environmental health hazards have failed to bring significant results in Poland—especially in the study region, because the major source of energy used for industrial production and house heating is coal combustion [3]. The principal reason for such behavior is energy poverty, and a low awareness of the risks persists in the population. A significant contribution to emissions and human exposure also comes from road transport, which has rapidly increased in the last decade in the study region [13].

Simultaneously, we observed that the highest number of outpatient visits and hospitalizations due to respiratory diseases (codes J00–J99 according to ICD-10) were evident during the coldest season. In the case of hospitalizations, the median number of patients during winter was approximately 2 times higher than in summer (136.0 vs. 79.5), and in the case of outpatient visits, the disparity was even higher (9664.5 vs. 2983). It is worth mentioning that forecasting, risk communication about the necessity of reducing emissions from individual heating stoves and medical services activation were the most effective in eliminating the harmful health consequences of winter smog conditions in the UK [11]. Available data suggest that improving air quality and reducing haze days in Beijing would benefit health, potentially resulting in reductions in the number of hospital emergency room visits due to respiratory diseases [14]. Current air quality in the central agglomeration area of Silesian Voivodeship is quite similar to those observed in China, probably because of the similarity of the sources of energy and heat production.

Another important issue in environmental epidemiology studies is the method of exposure estimation [15]. Usually, researchers assess the short-term health effects relating to increases in air pollutant concentration by the moving average concentration of pollution [16,17] or temporal lags (1–30 days) as a measure of exposure [18,19,20]. Commonly used models assess the risk ratio of health effects in response to an increase in pollutant concentration by a unit of a single IQR value. Our intention was, inter alia, to recognize which type of accepted method of the exposure presentation better reflects the concentration–response function in the Silesian province. The results from our study confirmed that the moving average concentration was better suited for the modelling of biological response as a result of PM_2.5_ or PM_10_ exposure. Evidently, each increase of dose expressed in the form of moving average concentration over a longer time (up to 2 weeks) leads to a slow increase in the daily number of outpatient visits or hospitalizations due to respiratory effects. The picture of the concentration–effect relationship assessed by the lag of health effect was not so clear. In the case of outpatient visits, the highest and statistically significant risk was observed with three or even five days lag after exposure for both pollutants, PM_10_ and PM_2.5_. In the case of hospitalizations, a significant health effect was noted only for two weeks lag of exposure. We observed an increase in health risk in relation to longer exposure times expressed by two or even four weeks moving average concentrations. The second way of exposure calculation (by temporal lag) also suggests an increase of outpatient visits and hospitalizations related to a longer time of exposure. It seems logical that, initially, people manage themselves in consultation with their doctor and in the absence of improvement they are hospitalized. Such an observation is in line with the well-known ‘air pollution health pyramid’ concept, where the effects have ranged from subtle subclinical effects through pharmaceutic usage and doctor consultancy to premature death [21]. It should be emphasized that the value of the estimated risk of respiratory hospitalization or outpatient visits related to an increase in particulate matter concentration depends on the method chosen to measure the exposure. However, it should be noted that regardless of the exposure presentation form, the statistically significant short-term effects of particulate matter exposure, manifested in ambulatory visits or hospital admissions due to the respiratory problem, was associated with a two-week exposure.

One limitation of this study is the secondary character of health data, which were obtained from the current system of National Health Found registration. It is impossible to rule out misdiagnosed cases, although in this paper we included only the number of diagnoses due to total respiratory diseases (codes J00–J99 according to ICD-10). Another issue is the applied exposure presentation as daily average values for the central agglomeration area without individual exposure data. However, this method, although imperfect, is often used in ecological epidemiological studies, mostly in regions where no individual PM_2.5_ or PM_10_ data are available. The use of one or several fixed monitoring stations is a common method of exposure estimation in short-term studies on health effects including mortality or morbidity [22,23]. Estimation of personal exposure might be better; however, the high cost of personal measurements makes it difficult to apply this approach for exposure estimation of a large population [24].

## 5. Conclusions

Summarizing the obtained results, it should be recognized that irrespective of the adopted exposure method (the moving average concentration or the temporal lag of the health effect in relation to the measured concentration), an increase of PM_2.5_ or PM_10_ by IQR leads to an increase of the number of respiratory problems among the population. The highest risk of hospitalization due to respiratory diseases was related to longer exposure expressed by two to four weeks of exposure. On the other hand, outpatient visits were related to a shorter exposure duration of 3 days. The results we obtained confirmed that the worst respiratory health situation were related to winter smog episodes that exist in Silesian Voivodship each year. This is important information for public health experts, and it should be taken into account in the planning of energy policies in Poland, especially in the Silesia province, which is the host of a current climate summit COP24 [25]. It certainly seems that a very high priority should be to continue efforts to educate the local population regarding the serious hazards to health from burning poor quality coal, biomass, or even garbage. We believe that newer stoves would yield better burning with lower gaseous and particulate pollutants emission. However, local government authorities should continue to support activities through co-financing modernization, at least for the poorest inhabitants.

## Figures and Tables

**Figure 1 ijerph-16-01131-f001:**
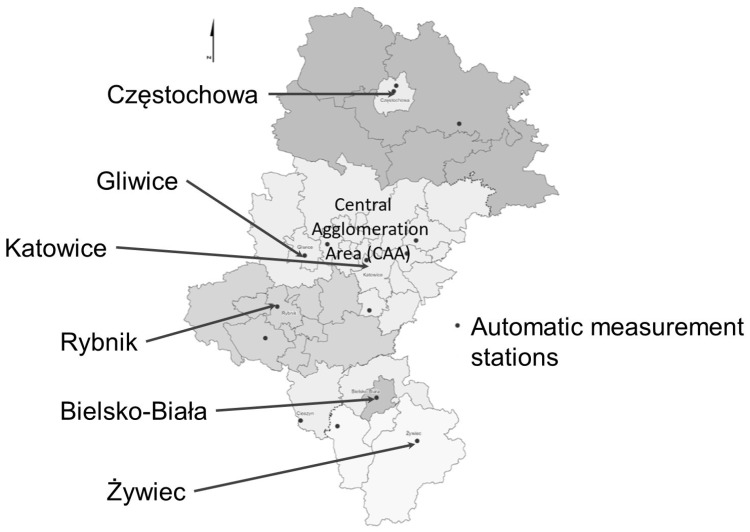
Map of the study region—Central Agglomeration Area (CAA); data according to the Regional Inspectorate for Environmental Protection in Katowice were available at the previous prognosis system’s website: http://spjp.katowice.wios.gov.pl/strefy2.html (cited 1 December 2018).

**Figure 2 ijerph-16-01131-f002:**
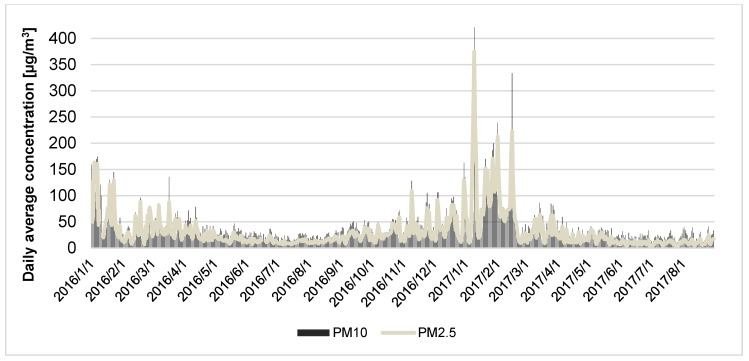
The daily average fine particulate matter concentration in the Silesia Agglomeration in the study period from 1 January 2016 to 31 August 2017.

**Figure 3 ijerph-16-01131-f003:**
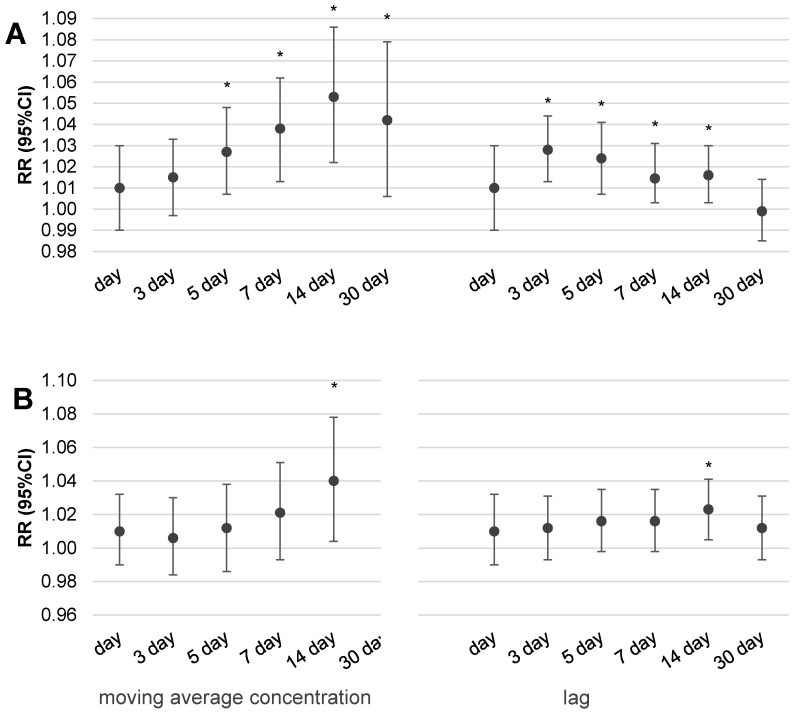
The risk ratio of daily outpatient visits (**A**) or hospitalizations (**B**) due to respiratory disorders related with the increase of PM_10_ concentration by IQR value (24.5 µg/m^3^) in two scenarios of exposure: expressed by the moving average concentration or related with a lag of health effect (RR was adjusted for season of the year, relative humidity, temperature, and atmospheric pressure; moreover, in the case of outpatient visits the model also included day of week; * indicates that a value is statistically significant). Legend: RR—risk ratio; CI—confidence interval; lag—temporal delay.

**Figure 4 ijerph-16-01131-f004:**
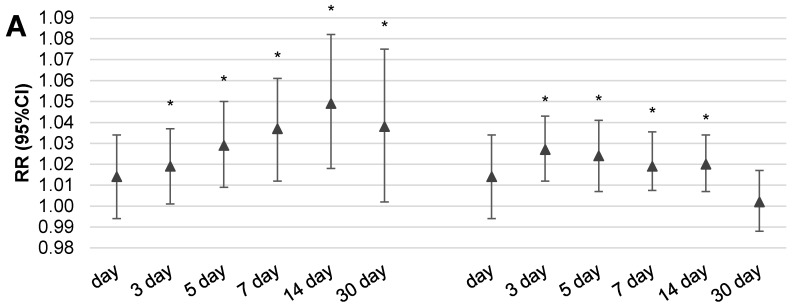
The risk ratio of daily outpatient visits (**A**) or hospitalizations (**B**) due to respiratory disorders related with the increase of PM_2.5_ concentration by IQR value (22.5 µg/m^3^) in two scenarios of exposure: expressed by the moving average concentration or related with a lag of health effect (RR was adjusted for season of the year, relative humidity, temperature, and atmospheric pressure; moreover, in the case of outpatient visits the model also included day of week; * indicates that a value is statistically significant). Legend: RR—risk ratio; CI—confidence interval; lag—temporal delay

**Table 1 ijerph-16-01131-t001:** The average daily concentration of ambient air pollution and respiratory health outcomes by season of the year in the study period from 1 January 2016 to 31 August 2017.

Variable (µg/m^3^)	Median Value (Interquartile Range (IQR)) in Particular Astronomical Season for Both Study Years (1 January 2016 to 31 August 2017)				
	Total	Winter 21 Dec–19 Mar	Spring 20 Mar–20 Jun	Summer 21 Jun –21 Sep	Autumn 22 Sep–20 Dec
SO_2_	8.36 (9.6)	18.2 (17.6)	7.3 (4.8)	4.7 (1.8)	9.3 (9.2)
NO_2_	22.67 (12.7)	28.7 (16.9)	20.8 (10.2)	17.7 (7.5)	26.2 (13.5)
NO_X_	30.83 (26.2)	41.0 (46.9)	27.2 (20.5)	23.9 (13.2)	40.7 (36.2)
PM_10_	30.33 (24.5)	52.6 (53.2)	28.5 (14.7)	20.1 (8.7)	34.3 (22.7)
PM_2.5_	19.50 (22.5)	44.0 (49.0)	17.0 (12.5)	11.0 (4.5)	25.0 (19.5)
Respiratory health outcomes (ICD-10: J00–J99)					
Hospitalization	100.0 (62.0)	136.0 (69.5)	96.0 (62.0)	79.5 (57.0)	94.0 (55.0)
Outpatient visits	4927.0 (7164)	9664.5 (10,746.5)	5436.5 (6308.0)	2983.0 (3252.5)	7516.0 (7414)

**Table 2 ijerph-16-01131-t002:** Values of Spearman correlation coefficients in the relationships between the concentrations of fine particulate matter and other ambient air pollution and meteorological conditions (Silesia Agglomeration in the period from 1 January 2016 to 31 August 2017).

Variable	Spearman Correlation Coefficient	
	PM_10_ (µg/m^3^)	PM_2.5_ (µg/m^3^)
SO_2_ (µg/m^3^)	0.78	0.82
NO_2_ (µg/m^3^)	0.84	0.76
NO_x_ (µg/m^3^)	0.82	0.76
O_3_ (µg/m^3^)	−0.56	−0.69
CO (µg/m^3^)	0.77	0.81
Atmospheric pressure (mmHg)	0.20	0.25
Temperature (°C)	−0.52	−0.68
Relative humidity (%)	0.19	0.36
Wind speed (m/s)	−0.19	−0.11

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
