# Peer review of "The Relationship between Daily Concentration of Fine Particulate Matter in Ambient Air and Exacerbation of Respiratory Diseases in Silesian Agglomeration, Poland"

_ijerph, 2019, doi:10.3390/ijerph16071131_

Round 1
Reviewer 1 Report
This research goal is an important one. We need to better correlate air pollution events with health outcomes. However, there are a number of concerns about this analysis:
(i) are the results statistically significant? In the graphs of moving averages, the error bars for all of the days overlap, suggesting that there are no significant differences. More discussion about the interpretation of these graphs is necessary. What information can truly be discerned from these overlapping error bars?
(ii) we know that air pollution levels vary across even short distances, so the two locations for air quality monitors may not accurately represent the entire region; there needs to be more justification for how these two stations can be used as representative of such a large area.
(iii) It is unclear how the “expected number of visits to the hospital” is estimated for the Poisson Distribution. The shortlist of confounding factors is included, but once those factors are used in the analysis, how well is the “expected number of visits” truly estimated?
(iv) what is the takeaway lesson from Table 1? The concentrations in the table are not particularly high and certain include wide variability. What is the utility of this information?
(v) what is the method for calculating RR? how many hospitalizations are included in the analysis (136 and 79---are these correct numbers? where are they included in the results?)? The number of hospitalizations seems small for drawing conclusions. There needs to be more explanation of the statistical analysis that allows this conclusion to be made.
(vi) the discussion needs to acknowledge more of the limitations in this data.
Minor comments:
There are a number of grammatical errors and incorrect word choices that should be addressed by having the paper edited by a native English speaker.
Figure 3 is repeated twice---should there be a Figure 4?
For Fig 3 and 4, be sure to include the acronyms in the legend.
Author Response
We really appreciate for the effort given during reviewing and insightful reading of our manuscript. All of the excellent comments we tried to incorporate into the text and answer the reviewer questions to explain any misunderstandings. We believe that the changes given in the reviewed version of the manuscript are adequate.

Reviewer 2 Report
The manuscript contains interesting data. It includes a satisfactory and properly selected list of relevant references. I suggest that it should be accepted for publication after some minor revisions described below.
1. First of all, there is no clear and concise statement as to why the study was performed and how readers will benefit from the results. It should be highlighted.
2. I believe the first part of the first sentence of the abstract ("worsening of air quality during colder season of the year") is not true for many areas in the world. Please specify it.
3. The statement: "significant worsening of ambient air quality during cold season" (line 23) is too general.
4. I think you should change the keyword " ecological study".
5. Please give the source of the statement in the first sentence of Introduction section.
6. Line 77, please specify: daily air temperature, relative air humidity
7. Line 80: It should be "less or equal".
8. Lines 82-82: I think you should describe the relationship of the parameters more precisely. Especially you can give us number of data, regression equation, etc.
9. Lines 89-90: Please justify.
10. Line 112: I think you mean: " daily mean concentrations".
Author Response
We really appreciate for the effort given during reviewing and insightful reading of our manuscript. All of the excellent comments we tried to incorporate into the text and answer the reviewer questions to explain any misunderstandings. We believe that the changes given in the reviewed version of the manuscript are adequate.
Comments and Suggestions for Authors
The manuscript contains interesting data. It includes a satisfactory and properly selected list of relevant references. I suggest that it should be accepted for publication after some minor revisions described below.
1. First of all, there is no clear and concise statement as to why the study was performed and how readers will benefit from the results. It should be highlighted.
Thank you for this comment. In the line 42-44 it was written: “The obtained results are necessary for the environmental health risk communication and will prove to be useful indicator for the implementation of the necessary health policy actions in the region.” And in the line 261-263 we explained, that : “These are important information for public health experts, and it should be taken into account in the planning of energy policy in Poland, especially in the Silesia province, which is the host of current climate summit COP24”. Perhaps this is not enough, so according to the reviewer's expectations we have added the following explanation in the text of Introduction section (kindly see below):
“Episodes of winter smog in Poland occur every year, always in the cold season. The level of environmental health perception in the country belongs to one of the lowest in Europe, however contemporary media publicize the problem and people panic. The situation is particularly difficult in the Silesia, a region where coal is mined and many people are employed in the mining industry. Our task is to document the relationship between the emerging hazard and the health of the population and communicate the risk based on reliable evidence. We believe that results of our study will help to improve the inhabitants' knowledge on the real hazard, and will force the social activity to improve the quality of environment”
2. I believe the first part of the first sentence of the abstract ("worsening of air quality during colder season of the year") is not true for many areas in the world. Please specify it.
Thank you for the comment, we agree that those problem affects some countries, usually with cold climate (like Canada, China, Poland) in which energy policy is based on coal combustion. So we correct the sentence, kindly see below:
“Relationship between worsening of air quality during colder season of the year and respiratory health problems among the exposed population in many countries located in cold climate was well documented in numerous studies.”
3. The statement: "significant worsening of ambient air quality during cold season" (line 23) is too general.
You are right, so we modified the sentence to the following:
“The obtained results confirmed significant increase of daily fine particulate matter concentration in ambient air during cold season in the study region with simultaneously increase of the number of outpatient visits and hospitalizations due to respiratory diseases”
4. I think you should change the keyword " ecological study".
We removed the keyword “ecological study”
5. Please give the source of the statement in the first sentence of Introduction section.
We confirmed given information by proper bibliography [1. Air quality monitoring system. State Inspectorate of Environmental Protection in Katowice, 2017 [quoted on 11 January 2018]. Available: http://powietrze.katowice.wios.gov.pl/dane-pomiarowe/automatyczne [in Polish] and we had renumerated the literature.
6. Line 77, please specify: daily air temperature, relative air humidity
It was done, thank You for the comment.
7. Line 80: It should be "less or equal".
We agree for the comment, we added “or equal”.
8. Lines 82-82: I think you should describe the relationship of the parameters more precisely. Especially you can give us number of data, regression equation, etc.
We explain that in whole study period we have: 3,550,901 outpatients visits and 60,346 hospitalizations due to respiratory diseases. Table 1 presents just mean value of daily hospitalizations or daily outpatient visits in particular seasons of the year (winter, spring, summer or autumn).
We rearranged a bit the sentence , in which we explain the first formula describing regression equation:
“…X is a vector of independent variables (daily concentration of particulate matter, daily meteorological conditions, season of the year, influenza episodes, and weekend days) and β means the calculated regression coefficient.”
9. Lines 89-90: Please justify.
We decided to use the log-linear Poisson regression because many published articles suggest that authors used this method in the case of small region with low level of daily variability of territorial PM concentrations (high value of correlation coefficient between measurement stations in the CAA). Moreover our earlier results assessing respiratory mortality in relation to particulate matter were also used this model.
10. Line 112: I think you mean: " daily mean concentrations".
We corrected text according to reviewer suggestion
Reviewer 3 Report
This paper could have a substantial positive impact for policy relating to public health.
The statistical results of the study are far from surprising.
Perhaps the most important aspect of the entire study is noted in the discussion, in lines 193 to 196. Yet, this is not reflected in the Conclusions section.
It certainly seems that a very high priority should be continued efforts to educate the local populace regarding the serious hazards to health from "burning poor quality coal, biomass, or even garbage."
If the biomass used is largely wood, then probably better-ventilated, or newer, stoves would yield better burning of the wood with less CO or other noxious gas products, and probably yield more heat output from the wood used.
I believe that a re-iteration of the points in lines 193-196 should be included and emphasized in the conclusions, You should stress a policy implication of persistence in educating the public regarding environmental health, for local authorities in Silesia as well as in other similar areas in cold climates.
I have made numerous English language edits, and some corrections in the text. Please see pdfs of each page, 1 through 9.
I list here a number of questions and comments relating to the text as written.
In line 83 (p. 3), three correlations are listed. It is not clear as written to which pair of variables each correlation refers.
In line 85, you use the word 'convergence' but perhaps mean correlation.
The long sentence (fragment) starting with "Because" in line 82 is a subordinate clause and not a sentence as written; also, the verb is "were used" but there is no subject - even as a clause.
In the caption to Figure 3 (PM10), the wording at line 154 would be better if it were written:
... ; moreover, in the case of outpatient visits, the model also included day of week

Author Response
We really appreciate for the effort given during reviewing and insightful reading of our manuscript. All of the excellent comments we tried to incorporate into the text and answer the reviewer questions to explain any misunderstandings. We believe that the changes given in the reviewed version of the manuscript are adequate.
Comments and Suggestions for Authors
This paper could have a substantial positive impact for policy relating to public health.
The statistical results of the study are far from surprising.
Perhaps the most important aspect of the entire study is noted in the discussion, in lines 193 to 196. Yet, this is not reflected in the Conclusions section.
It certainly seems that a very high priority should be continued efforts to educate the local populace regarding the serious hazards to health from "burning poor quality coal, biomass, or even garbage."
If the biomass used is largely wood, then probably better-ventilated, or newer, stoves would yield better burning of the wood with less CO or other noxious gas products, and probably yield more heat output from the wood used.
I believe that a re-iteration of the points in lines 193-196 should be included and emphasized in the conclusions, You should stress a policy implication of persistence in educating the public regarding environmental health, for local authorities in Silesia as well as in other similar areas in cold climates.
Thank you so much for all comments. According to reviewer expectation we include the following sentences in the Conclusion:
“It certainly seems that a very high priority should be to continue efforts to educate the local population regarding the serious hazards to health from burning poor quality coal, biomass, or even garbage. We believed that newer stoves would yield better burning with lower emission gaseous and particulate pollutants. However, local government authorities should continue supporting activities through co-financing modernization, at least for the poorest inhabitants.”
I have made numerous English language edits, and some corrections in the text. Please see pdfs of each page, 1 through 9.
Thank You so much for very insightful correction English, it was done (kindly see the current version of manuscript)
I list here a number of questions and comments relating to the text as written.
In line 83 (p. 3), three correlations are listed. It is not clear as written to which pair of variables each correlation refers.
We explain that the following correlation coefficients describe relationship between:
PM2.5 concentration measured in both stations (Gliwice and Katowice) R=0.93
SO2 concentration measured in both stations (Gliwice and Katowice) R= 0.90
Ambient air temperature measured in both stations (Gliwice and Katowice) R= 0.99
Relative humidity of air measured in both stations (Gliwice and Katowice) R=0.99
In line 85, you use the word 'convergence' but perhaps mean correlation.
We changed “convergence” to “correlation”.
The long sentence (fragment) starting with "Because" in line 82 is a subordinate clause and not a sentence as written; also, the verb is "were used" but there is no subject - even as a clause.
You are right, we rearranged a bit the sentence and we belief that now is more clear (kindly see below):
“Because of very high correlation coefficients between daily averages recorded at both stations (r = 0.93; r = 0.90 and r = 0.99, for PM2.5, SO2 and ambient air daily air temperature or relative air humidity, respectively) values from Katowice station were used in the further analysis the measurement.”
In the caption to Figure 3 (PM10), the wording at line 154 would be better if it were written:
... ; moreover, in the case of outpatient visits, the model also included day of week
It was done, kindly see the current version.
Reviewer 4 Report
As a reviewer I have the following remarks:
Major
Lines 59-69. The registered outpatient visits = emergency visits? Is it possible that a person was used twice; as outpatient and again as hospital? How do you assign persons to the considered location, have you used home addresses? It should be clarified.
Your model (formula (1) –you don’t have time and adjustment for it. Why you don’t use GAM or case-crossover method? We see more cases in winter, but it’s not necessary due to air pollution. This should be explained.
Usually in such study (respiratory) flu indicator is used for days with even one flu case.
Minor
Line 20. Suggestion “particulate matters (PM: PM2.5 and PM10)” or full definition.
Line 47: “increasing drug”?
Line 52: You have “dose-response”, I suggest (in whole text) to use “concentration-response”.
Figure 1. Even with Paint it’s possible to write in large fronts names of the cities (now almost invisible).
Line 82. Could you please provide the distance between two stations.
Line 85. “convergence”?
Line 100, formula (2), in my opinion should be, RR=exp(b*delta)?
Line111: please keep 1 digits after dot: 44.0, 52.6 etc.
Figure 2, is it possible to show only up to 420 or 430. Now we have empty space above (it’s only suggestion).
Table 1. Do we really need 2 digits accuracy? Also Winter 21.12-19.3 in my opinon looks better, also agrees with your dates.
Table 2. I suggest to remove p-values. You may say on it. Try to reduce.
Figure 3 and 4 (you have 3 and 3).
Figure 3 and 4. Replace LCI-UCI (if so should be Limit, LCIL? )I suggest RR (95% CI).
Line 153: RR adjusted for season. See my Minor question 2.
Figure 3 and 4 You can use Paint, cut PM10, put it on the graph, increase the presented area. Show more what is more important. It’s only suggestion.
Please read again your paper, you have somewhere PM abbreviation –not defined.
Why do you mentioned other air pollutant but only PMs were used?
Thank you
Author Response
We really appreciate for the effort given during reviewing and insightful reading of our manuscript. All of the excellent comments we tried to incorporate into the text and answer the reviewer questions to explain any misunderstandings. We believe that the changes given in the reviewed version of the manuscript are adequate.
Comments and Suggestions for Authors
As a reviewer I have the following remarks:
Major
Lines 59-69. The registered outpatient visits = emergency visits? Is it possible that a person was used twice; as outpatient and again as hospital? How do you assign persons to the considered location, have you used home addresses? It should be clarified.
Thank you for the comment. We explain, that we got data of registered outpatient visits and we have no information if they are emergency visits. We can't excluded that one person was used twice: as outpatient and again as hospital admission. Unlikely on the same day, but we do not know that. Anyway, we have focused on daily number of events, not number of patients (individuals).
In the registry we have the identification number of place of residence to separate communes or cities – local administrative units according to NTS-4 (Nomenclature of Territorial Units for Statistics). In current paper we present the total number of health events in the entire Central Agglomeration Area, which we defined in Material and methods section (a region of the Upper Silesia Agglomeration and the central part of the Silesian zone with the following poviats: Będzin, Zawiercie, Tarnowskie Góry, Gliwice and Bieruń-Lędziny Counties).
Your model (formula (1) –you don’t have time and adjustment for it. Why you don’t use GAM or case-crossover method? We see more cases in winter, but it’s not necessary due to air pollution. This should be explained.
We decided to use the log-linear Poisson regression because many published articles suggest that authors used this method in the case of small region with low level of daily variability of territorial PM concentrations (high value of correlation coefficient between measurement stations in the CAA). Moreover our earlier results assessing respiratory mortality in relation to particulate matter were also used this model. We know that some authors used crossover method, other used GAM. We present just preliminary results of our study (with seasonality effect) and we plan to use above mentioned methods to analyse data from entire voivodeship.
We know, that infectious diseases (especially influenza) rapidly increase with lowering temperature of ambient air in colder season so we controlled this variable in the model as dichotomous variable (influenza yes or not). We consulted decision with the sanitary inspectorate in the Silesian voivodehip, which suggest that in the entire study period we had no influenza epidemic. On the second hand publications on quality of air in the study region suggest that individual heating remains a major sources of particulate matter emission (Kowalska 2011 and Rogula-Kozłowska 2013).
Usually in such study (respiratory) flu indicator is used for days with even one flu case.
This is valuable advice. We will use your suggestion in future analyses for the entire voivodeship
Minor
Line 20. Suggestion “particulate matters (PM: PM2.5 and PM10)” or full definition.
It was done.
Line 47: “increasing drug”?
Thank you for the comment, of course you are right. We correct the text, kindly see the following text: “increasing of medicament using”
Line 52: You have “dose-response”, I suggest (in whole text) to use “concentration-response”.
We agree with the reviewer, it was done in whole text.
Figure 1. Even with Paint it’s possible to write in large fronts names of the cities (now almost invisible).
It was done, we hope that current version of the Figure 1 will be accepted
Line 82. Could you please provide the distance between two stations.
The distance between Katowice and Gliwice measurement stations is about 25 km
Line 85. “convergence”?
We have corrected “convergence” into ‘correlation”
Line 100, formula (2), in my opinion should be, RR=exp(b*delta)?
You are right, we corrected the formula
Line111: please keep 1 digits after dot: 44.0, 52.6 etc.
It was done
Figure 2, is it possible to show only up to 420 or 430. Now we have empty space above (it’s only suggestion).
It was done
Table 1. Do we really need 2 digits accuracy? Also Winter 21.12-19.3 in my opinon looks better, also agrees with your dates.
OK, we change the text according to reviewer suggestion
Table 2. I suggest to remove p-values. You may say on it. Try to reduce.
It was done
Figure 3 and 4 (you have 3 and 3).
You are right, we have fixed the mistake.
Figure 3 and 4. Replace LCI-UCI (if so should be Limit, LCIL? )I suggest RR (95% CI).
It was done
Line 153: RR adjusted for season. See my Minor question 2.
Figure 3 and 4 You can use Paint, cut PM10, put it on the graph, increase the presented area. Show more what is more important. It’s only suggestion.
It was done.
Please read again your paper, you have somewhere PM abbreviation –not defined.
We explained the PM abbreviation
Why do you mentioned other air pollutant but only PMs were used?
We assure, that we have assessed dependencies for the all remaining pollutants (also gaseous), but the results are so many that we decided to limit it only to dust in the current paper. Final report is too large it takes hundred pages, if you will be interested we can send it.
Round 2
Reviewer 1 Report
I think this research is important and relevant publish, however, I have two remaining issues with the paper:
Table 2 describes the Spearman Correlation between air pollution and meteorological conditions. What is the data that is used for the meteorological conditions (temperature? wind? inversion layer?....). Please add more detail.
Figure 4 shows the relative risk values (and, thank you for adding the stars for statistical significance). The RR values are small and not all are significant, and this is not uncommon for air pollution correlations with general health conditions. These are difficult correlations since there are so many confounding factors. This lack of strong correlation is not explained well in the discussion. It is important to include both the conclusions, but also the limitations of research findings. This needs better balance.
Author Response
Again we very thank for the revision. We assure, that our work has been corrected by a native speaker. We believe that the changes given in the reviewed version of the manuscript and our explanations are adequate.
Table 2 describes the Spearman Correlation between air pollution and meteorological conditions. What is the data that is used for the meteorological conditions (temperature? wind? inversion layer?....). Please add more detail.
We kindly explain, that 24-hour regional monitoring in Silesia region (such as in entire country) include only the following data of meteorology:
Atmospheric pressure [mmHg]
Temperature [˚C]
Relative humidity [%]
direction [0 ] and speed [m/s] of the wind.
According to Rewiever expectation we added the following sentence ‘ Moreover, the dominant direction of the wind was the west direction (49.5%), next south (20.5%) and east (20.5%) direction.” (kindly see the line 153-154) and also we included values of correlation coefficient between particulate matter concentration and speed of the wind, kindly see the Table 2.
Figure 4 shows the relative risk values (and, thank you for adding the stars for statistical significance). The RR values are small and not all are significant, and this is not uncommon for air pollution correlations with general health conditions. These are difficult correlations since there are so many confounding factors. This lack of strong correlation is not explained well in the discussion. It is important to include both the conclusions, but also the limitations of research findings. This needs better balance.
Thank You for this comment, you are right. Obtained results present not large RR values however statistically significant. According your suggestion we added the following explanation with hope that current version will be acceptable: “In conclusion we have to underline that, the obtained risk ratios were small, increase of PM10 or PM2.5 concentration by IQR value increased the risk of outpatient visits or hospitalization due to respiratory diseases, average by 2-5%." (kindly see the line 201-204).